# GRADIENT-CONSTRAINED TRAINING FOR DISTRIBUTED LARGE LANGUAGE MODELS

## ABSTRACT

Computational constraints make distributed architectures essential for working with large-Language models (LLMs), while inter-node gradient synchronization often becomes a major bottleneck in the distributed parallel training. Current compression techniques mainly aim to reduce communication volume for the computed gradients, instead of generating gradients with inherent sparsity directly during training. In this paper, we propose gradient constrained training (GCT), a novel approach that leverages gradient constraints to generate low-rate gradients. By balancing performance and rate, we directly form an effectively training-time gradient source, achieving high compression efficiency with no accuracy degradation. In extensive experiments, we observed that GCT provides at least 70% average bitrate savings and demonstrates consistent and stable improvements in coding efficiency across various model tasks and distributed systems, which indicates that GCT have profound implications for next-generation distributed model training and stable gradient transmission.

## 1 INTRODUCTION

Distributed training with data, model, and pipeline parallelism has become essential for pre-training and fine-tuning modern LLMs, as single-GPU setups are infeasible given dataset and model scale. However, gradient storage, communication, and synchronization in such systems impose substantial memory and bandwidth overhead, often leading to high latency and bottlenecks that limit training efficiency and scalability. Gradient compression has thus emerged as a key technique for enabling efficient large-scale distributed training.

Existing gradient compression techniques can be broadly categorized into sparsification or quantization methods. The core idea of sparsification is to transmit only the most important subset of gradient values while discarding or accumulating the remaining values for later transmission (Aji & Heafield, 2017; Stich et al., 2018; Basu et al., 2019). These sparsification practices have shown that, the communication of most parameter gradients make less contribution for parameter updates. The essence of quantization is to represent gradients using lower-precision numerical values, thus reducing the number of bits required per gradient value (Seide et al., 2014; Alistarh et al., 2017; Wen et al., 2017). Both of them fundamentally leverage the inherent sparsity of natural gradient distributions by applying numerical processing to the gradient data after it is generated, instead of considering directly produce a more encoding-friendly gradient source.

BackSlash (Wu et al., 2025) pioneered the introduction of rate-distortion theory (Berger, 2003) into the field of model compression in training. By adding constraint to the loss function, they achieved deep compression of parameters during the training process, demonstrating effective parameter compression in most cases, which inspired our further exploration of in-training gradient compression. We aim to employ a similar approach to deeply optimize the gradient distribution generated during model training, enabling superior compression performance in subsequent quantization, sparsification, or entropy encoding processes.

In this paper, we attempt to integrate the gradient constraint to the loss function and train the LLMs in this gradient-constrained paradigm. By balancing model performance and gradient bitrate, it simultaneously reduces the model generalization error and gradient structural error in training and generate compression-friendly gradient streams, which can notably enhance the performance of subsequent compression algorithms. Our main contributions can be noted as follows:

1. We observed that the gradients exhibits strong characteristics of a generalized Gaussian distribution. It led us to develop a distributed training framework that employs exponential Golomb (EG) coding for gradient-encoded communication. The framework significantly reduces gradient communication volume, effectively alleviating the bandwidth bottleneck in distributed training.

2. We propose a novel gradient-constrained training method. By introducing gradient constraints, we achieved compression of model training gradients for the first time. Extensive experiments show that the gradients obtained with this training paradigm exhibit a sparser distribution, along with smaller and more stable encoding results.

3. We developed an efficient and low-overhead strategy for approximating second-order derivatives. In practice, this estimation method requires only storing the previous step's gradients to compute second-order derivatives, with almost no increase in training time. It significantly reduces the computational and spatial demands of second-order derivative backpropagation, enhancing the practicality of gradient-constrained training algorithms.

## 2 RELATED WORK

### 2.1 GRADIENT QUANTIZATION

One method to compress gradient information is gradient quantization, whose core idea is to quantize high-precision gradient data into low-precision values to alleviate communication bottlenecks.

Early research focused on the number of quantization bits. Seide et al. (2014) were the first to propose quantizing gradients to 1-bit to reduce communication volume between models, though this approach could slow down training convergence. To improve model convergence, TernGrad (Wen et al., 2017) introduced a ternary quantization method combining gradient clipping and quantization.

Quantization-induced training errors may accumulate and amplify during continued training, leading to the development of error accumulation methods in gradient quantization. The QSGD scheme proposed by Alistarh et al. (2017) provided a theoretically provable stochastic and unbiased quantization function for gradient data, incorporating quantization compensation to mitigate the negative effects of quantization errors. Wu et al. (2018) proposed an error-compensated quantized stochastic gradient descent algorithm, which reduces the error bound by accumulating all historical quantization errors.

Sign-based quantization algorithms have also continued to evolve. signSGD, introduced by Bernstein et al. (2018), adopted an aggressive gradient compression strategy in which each worker node only transmits the sign of each mini-batch stochastic gradient. Karimireddy et al. (2019) addressed the bias issue in the signSGD quantization estimator through an error feedback mechanism, resolving its tendency to diverge from the optimal solution during training and enabling stable convergence across various model trainings.

Additionally, some studies have explored adaptive quantization level adjustment. For example, Faghri et al. (2020) approached from the perspective of information rate-distortion theory and proposed the AGQ algorithm, which computes near-optimal quantization levels in an online and low-cost manner. However, gradient quantization requires reducing the number of bits for transmitting gradients, which inherently faces limitations due to the loss of gradient precision.

### 2.2 GRADIENT SPARSIFICATION

Compared to gradient quantization, sparsification can achieve compression benefits that are orders of magnitude higher. Sparsification operates on the premise that not all gradient components contribute equally to parameter updates; thus, in each transmission cycle, only a subset of gradients with higher importance is transmitted.

Active work in this area employs Top-k sparsifiers, which select the k largest gradients for communication (Aji & Heafield, 2017; Basu et al., 2019) while directly discarding smaller gradients. Stich et al. (2018) analyzed a method that randomly selects k components, effectively treating all gradients equally. Song et al. (2021) combined these approaches by proposing a Bayesian prior-based gradient sampling method, achieving adaptive gradient sparsification. More recently, Wang et al.

(2025) developed a gradient sparsification enhancement method inspired by neural network pruning. Lin et al. (2018) proposed the DGC algorithm, which achieves sparsity during communication by locally accumulating smaller gradients instead of transmitting them immediately.

Further research has explored the combination of sparsification and quantization. For example, Jiang & Agrawal (2018) introduced PQASGD, a method that first sparsifies the gradients and then applies product quantization to the remaining sparse gradients. Yan et al. (2022) established a systematic approach to dynamically adjust quantization precision and sparsification parameters, addressing the manual tuning requirement in PQASGD. Their method continuously optimizes these parameters based on a comprehensive analysis of gradient vector norms, predefined communication resource allocation, and quantifiable residual iteration counts within the training framework. Additionally, Xie et al. (2024) conceptualized sparsification as 0-bit quantization, reformulating the problem as a mixed-precision quantization task and thereby integrating the two-stage approach into a unified framework.

Essentially, both quantization and sparsification compress gradients at the data level by exploiting their inherent numerical redundancy. However, they do not actively induce a sparser distribution in the gradients themselves.

## 3 GENERALIZED GAUSSIAN DISTRIBUTED TRAINING ARCHITECTURE

### 3.1 GENERALIZED GAUSSIAN DISTRIBUTION OF GRADIENTS

Prior distributions for model parameters (e.g., Gaussian, Laplace) have been extensively studied and utilized in tasks like parameter initialization and model compression. In contrast, gradient distributions during training have been less explored, largely due to their stochastic nature, which makes them difficult to track and regulate. While gradient distributions are often assumed to be Gaussian in classical stochastic optimization for mathematical convenience, empirical evidence in deep learning consistently shows that they exhibit significant heavy-tailed characteristics, strongly deviating from Gaussian assumptions.

Our systematic analysis reveals that gradients across various models and training stages consistently exhibit peaky, heavy-tailed distributions, well-characterized by generalized Gaussian distributions with low shape parameters. For instance, during BERT training (Devlin et al., 2019), gradients sampled at the end of epochs 1, 5, and 10 yield shape parameters of 0.14, 0.27, and 0.32, respectively (Fig. 1), confirming their close fit to a generalized Gaussian model with low shape parameters.

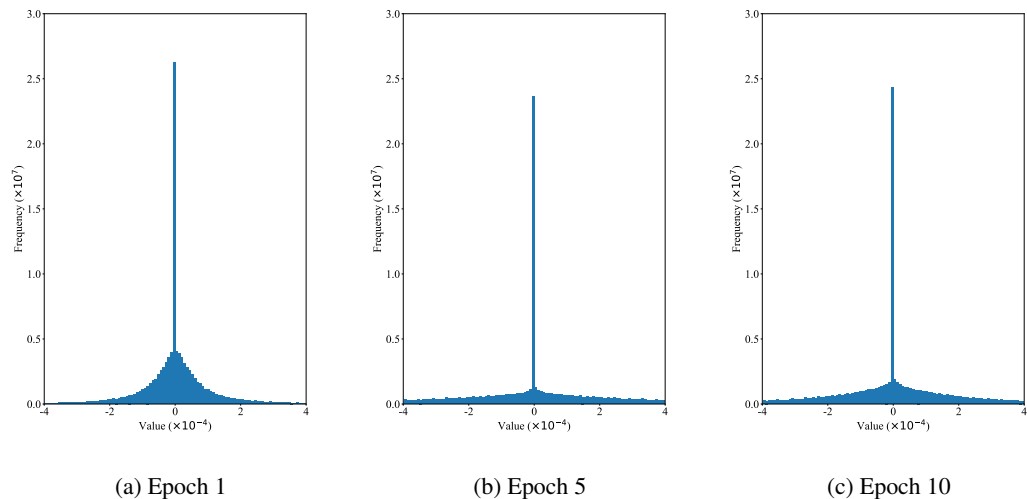

|                 |                 |                  |
| :-------------: | :-------------: | :--------------: |
|   (a) Epoch 1   |   (b) Epoch 5   |   (c) Epoch 10   |

Figure 1: The gradient distribution collected at different stages when training BERT on the IMDB dataset.

Research by Wen & Villasenor (1999) pointed out that Exponential Golomb coding offers high coding efficiency for generalized Gaussian sources. Given that gradient distributions conform to the characteristics of generalized Gaussian distributions, we are inspired to introduce Exponential Golomb coding into the distributed training framework for encoding gradients transmitted between models. Based on this idea, we will design a distributed hardware training framework based on Exponential Golomb coding.

### 3.2 EXPONENTIAL GOLOMB CODING FOR GRADIENT COMPRESSION

Exponential-Golomb (EG) coding is a lossless compression technique for encoding sequences of positive integers into a compact bitstream. It achieves optimal compression efficiency—reaching the entropy lower bound—when the input symbols follow a geometric distribution. Prior studies, such as Wen & Villasenor (1999), have extended its evaluation to more general source models, including the family of Generalized Gaussian distributions.

Compared to traditional Huffman coding, Exponential-Golomb (EG) coding eliminates the need to construct and transmit code tables, reducing both storage and communication overhead. Moreover, EG coding is robust to shifts in distribution parameters, allowing it to adapt to dynamic layer-wise gradient variations during training. This adaptability makes EG coding particularly suitable for gradient compression in distributed training.

As demonstrated in Algorithm 1, after computing the local layer, the worker node first performs dynamic range normalization and quantization processing; after the parameter server receives the compressed layer, it performs EG decoding and inverse quantization, and then performs layer aggregation.

---

**Algorithm 1** Distributed Gradient Compression with Entropy Coding (Concise)

---

**Require:** Local gradients $\{\mathbf{g}_k\}_{k=1}^K$, quantization step $2^{-n}$, entropy encoder/decoder $(\mathrm{Enc}, \mathrm{Dec})$
  1: **WORKERS (IN PARALLEL):**
  2: $\mathbf{g}_k \leftarrow \nabla \mathcal{L}_k(\Theta)$
  3: $Q_k \leftarrow \mathrm{round}(2^n \mathbf{g}_k)$
  4: $H_k \leftarrow \mathrm{Histogram}(Q_k); \ C_k \leftarrow \mathrm{BuildCode}(H_k)$
  5: $B_k \leftarrow \mathrm{Enc}(Q_k; C_k); \ \textbf{SEND} \ (B_k, C_k)$ TO SERVER
  6: **SERVER:**
  7: **RECEIVE** $\{(B_k, C_k)\}_{k=1}^K$
  8: **for** $k = 1$ TO $K$ **do**
  9:     $Q_k \leftarrow \mathrm{Dec}(B_k; C_k)$
10:     $\mathbf{g}_k \leftarrow 2^{-n} Q_k$
11: **end for**
12: $\mathbf{g} \leftarrow \frac{1}{K} \sum_{k=1}^K \mathbf{g}_k$
13: $\Theta \leftarrow \Theta - \eta \mathbf{g}$
14: **BROADCAST** $\Theta$ TO ALL WORKERS
15: **RETURN** $\Theta$

---

### 3.3 DISTRIBUTED DATA PARALLEL

Distributed Data Parallel (DDP) (Li et al. (2020)), a distributed training framework implemented in PyTorch. During each training iteration, every node processes a distinct data subset, independently executes forward and backward passes. The resulting gradients are then synchronized through an all-reduce operation. Subsequently, each node performs identical parameter updates using the aggregated gradients, effectively emulating training with a larger batch size while distributing the computational load.

# 4 DECAYING GRADIENT-CONSTRAINED TRAINING (GCT)

## 4.1 GRADIENT RD COST

From the perspective of rate-distortion optimization (RDO) theory, constrained training can be viewed as an RDO process that seeks a balance between model performance and bit-rate. So we will progressively derive the gradient-constrained training algorithm from the viewpoint of RDO.

We construct the RD Cost based on the Lagrange multiplier method using the model loss and a gradient metric, which serves as the loss function for gradient constrained training. This approach aims to find the optimal balance between performance and bitrate during model training optimization:

$$\mathscr{L}(x, y; w) = D(x, y; w) + \lambda \cdot R(\frac{\partial D(x, y; w)}{\partial w}) \tag{1}$$

where $x$, $y$, and $w$ represent the input, label, and model weights, respectively. And $\lambda$ denotes the Lagrange multiplier.

$D(\cdot)$ represents the empirical loss of the model's fit to the data, which reflects the generalization error of the model on the dataset. $R(\cdot)$ represents the structural loss of the training gradients, which measures the bitrate of the gradients generated during training. The $\ell_1$ was selected for the $R(\cdot)$ implementation due to its closer alignment with the generalized Gaussian distribution of the gradients and the higher computational efficiency of its derivative.

Assuming the empirical loss function $D(\cdot)$ for the task is $\mathcal{L}_{\text{task}}$, the number of model parameters is $n$, and replacing the $R(\cdot)$ with $\ell_1$, the gradient RD cost to be optimized can be expressed as:

$$\mathscr{L}(x, y; w) = \mathcal{L}_{task}(x, y; w) + \lambda \cdot \sum_{i=1}^{n} |\frac{\partial \mathcal{L}_{task}(x, y; w)}{\partial w_i}| \tag{2}$$

## 4.2 WEIGHT UPDATE

Compared to the computation of the loss function, the back propagation and weight update require more complex discussion. If we directly aggregate the absolute values of all gradients and compute its gradient with respect to each parameter during backpropagation, then the update formula for parameter $w_i$ at step $t$ can be expressed as follows:

$$w_i^{(t+1)} = w_i^{(t)} - \eta \cdot \left( \frac{\partial \mathcal{L}_{task}(x, y; w)}{\partial w_i} + \lambda \cdot \text{Sign}(\frac{\partial \mathcal{L}_{task}(x, y; w)}{\partial w_i}) \cdot \sum_{j=1}^{n} \frac{\partial^2 \mathcal{L}_{task}(x, y; w)}{\partial w_j \partial w_i} \right) \tag{3}$$

Since $\frac{\partial^2 \mathcal{L}_{task}(x,y;w)}{\partial w_j \partial w_i}$ maybe depends on various parameters which may reside in different layers, we can only proceed with the second propagation after the backpropagation of the gradient $\frac{\partial \mathcal{L}_{task}(x,y;w)}{\partial w_i}$ has been completed. In this scenario, the weight update requires computing and storing the full Hessian matrix of the loss with respect to all parameters, which poses a catastrophic computational and storage burden.

A common approximation for the Hessian matrix is to retain only its diagonal elements. Specifically, we set $\frac{\partial^2 \mathcal{L}_{task}(x,y;w)}{\partial w_j \partial w_i} = 0$ when $j \neq i$. Then Equation 3 can be simplified as:

$$w_i^{(t+1)} = w_i^{(t)} - \eta \cdot \left( \frac{\partial \mathcal{L}_{task}(x, y; w)}{\partial w_i} + \lambda \cdot \text{Sign}(\frac{\partial \mathcal{L}_{task}(x, y; w)}{\partial w_i}) \cdot \frac{\partial^2 \mathcal{L}_{task}(x, y; w)}{\partial w_i^2} \right) \tag{4}$$

This technique can significantly reduce memory consumption, simplify the computational graph, and enable the parallel backpropagation of first and second-order derivatives.

## 4.3 APPROXIMATION OF SECOND-ORDER DERIVATIVES

Accurately calculating second-order derivatives is computationally prohibitive, as it requires a full backward pass—a cost unacceptable for billion-parameter LLMs. We therefore developed an estimation method that trades a tolerable precision loss for tractable computation.

To simplify notation, we denote the first- and second-order derivatives of parameter $w_i$ at step $t$ as $g_i^{(t)} = \frac{\partial L_{\text{task}}(x,y;w)}{\partial w_i}$ and $h_i^{(t)} = \frac{\partial^2 L_{\text{task}}(x,y;w)}{\partial w_i^2}$ respectively. In parameter update of GCT, the actual equivalent gradient should be $\hat{g}_i^{(t)} = g_i^{(t)} + \lambda \cdot h_i^{(t)}$, which replaces the original gradient $g_i^{(t)}$. Meanwhile, the learning rate $\eta$ is typically tiny in LLMs, so the weight change $\delta w = w_i^{(t)} - w_i^{(t-1)}$ is generally sufficiently small, allowing to approximate $\hat{h}_i^{(t)}$ using a quasi-Newton method to replace the true $h_i^{(t)}$:

$$\hat{h}_i^{(t)} = \frac{g_i^{(t)} - \hat{g}_i^{(t-1)}}{w_i^{(t)} - w_i^{(t-1)}} \tag{5}$$

Since $\hat{h}_i^{(t)}$ cannot be determined before estimating $\hat{g}_i^{(t)}$, we temporarily substitute $g_i^{(t)}$ for $\hat{g}_i^{(t)}$.

From the weight update for per step:

$$w_i^{(t)} - w_i^{(t-1)} = \eta \cdot \hat{g}_i^{(t-1)} \tag{6}$$

Substituting into Equation 5:

$$\hat{h}_i^{(t)} = -\frac{1}{\eta} \cdot \frac{g_i^{(t)}}{\hat{g}_i^{(t-1)}} + \frac{1}{\eta} \tag{7}$$

To prevent gradient explosion caused by excessively small gradients $\hat{g}_i^{(t-1)}$, we can introduce a small correction coefficient $\epsilon$ for gradient clipping. Then the Equation 7 can be modified as:

$$\hat{h}_i^{(t)} = -\frac{1}{\eta} \cdot \frac{g_i^{(t)}}{\hat{g}_i^{(t-1)} + \epsilon} + \frac{1}{\eta} \tag{8}$$

This method only requires the storage of equivalent gradients from the previous timestep to rapidly compute second-order derivatives for the current timestep. Although some estimation accuracy is sacrificed, it significantly optimizes storage and computational efficiency.

When using this approximation method, the weight update formula (4) in backpropagation can be simplified as:

$$w_i^{(t+1)} = w_i^{(t)} - \eta \cdot g_i^{(t)} - \lambda \cdot \text{Sign}(g_i^{(t)}) \cdot (1 - \frac{g_i^{(t)}}{\hat{g}_i^{(t-1)} + \epsilon}) \tag{9}$$

### 4.4 CONSTRAINT DECAY

As the model converges, the magnitude of gradients gradually approaches zero, while the second-order derivatives exhibit no significant or analyzable changes. Employing a fixed Lagrange multiplier would thus induce severe oscillations in the loss value during later training stages and significantly hinder convergence. To mitigate this, we introduce a decay coefficient $0 < \alpha < 1$, allowing the coefficient $\lambda$ to gradually diminish as training progresses. Specifically, upon the completion of the $\tau$-th training epoch, the multiplier is updated as:

$$\lambda_{\tau+1} = \alpha \cdot \lambda_\tau \tag{10}$$

This simple design ensures that, although the code length of the gradients may exhibit slight fluctuations in the later stages of training, loss value oscillations can be effectively resolved.

### 4.5 OVERALL OF GCT ALGORITHM

Based on the above discussion, we can summarize the decaying gradient-constrained training algorithm in a more engineering-oriented manner as follows:

---

**Algorithm 2** Decaying Gradient Constrained Training Algorithm (GCT)

---

1: **Require:** Empirical loss function $\mathcal{L}_{task}$, model weights $w$, training step $t = 1$, learning rate $\eta$, Lagrange multiplier $\lambda$, smoothing coefficient $\epsilon$, decaying coefficient $\alpha$.
2: **for** each epoch $\tau$ **do**
3:    **for** each batch $(x_b, y_b)$ **do**
4:       Forward and back propagation to compute the gradients: $g_i^{(t)} \leftarrow \frac{\partial \mathcal{L}_{task}(x_b, y_b; w^{(t)})}{\partial w_i^{(t)}}$.
5:       Compute approximate second-order derivatives: $\hat{h}_i^{(t)} \leftarrow (-\frac{1}{\eta} \cdot \frac{g_i^{(t)}}{\hat{g}_i^{(t-1)} + \epsilon} + \frac{1}{\eta}) \cdot \mathbf{1}_{t \geq 2}$.
6:       Compute and store the equivalent gradient: $\hat{g}_i^{(t)} \leftarrow g_i^{(t)} + \lambda_\tau \cdot h_i^{(t)}$.
7:       Weight update: $w_i^{(t+1)} \leftarrow w_i^{(t)} - \eta \cdot \hat{g}_i^{(t)}$, $t \leftarrow t + 1$.
8:    **end for**
9:    Constraint decay: $\lambda_{\tau+1} = \alpha \cdot \lambda_\tau$.
10: **end for**
11: **Until** convergence or max iterations.

---

## 5 EXPERIMENT

### 5.1 TRAINING PROCESS

To evaluate GCT on various tasks, we employ two examples: fine-tuning BERT on the IMDB dataset for classification, and DeepSeek on the SQuAD for generation. Throughout training, we sampled the loss and computed the average EG-encoded gradient length at regular intervals, as shown in Fig 2.

A comparative analysis of Fig 2a and Fig 2c reveals that the training curve of GCT is very stable. Compared to conventional training, GCT hardly affects the model's convergence speed. In the DeepSeek experiment, GCT converges even faster.

A comparative analysis of Fig 2b and Fig 2d indicates that, under the EG encoding, GCT achieves significantly better gradient compression efficiency throughout the entire training process compared to conventional training. Although the average code length variability increases during the later stages of BERT training due to the effects of constraint decay, overall, GCT maintains a more stable distribution of the gradient code length.

### 5.2 ABLATION

To investigate the impact of the gradient constraint on model training on later training stages, we will perform an ablation study on constraint decay. Specifically, we again use the BERT task as an example, comparing conventional training, decaying GCT ($\lambda = 0.8$), and maintained GCT ($\lambda = 1.0$). The detailed training processes are recorded in Fig 3.

Observing the training loss, both the decaying GCT and conventional training steadily converge. However, the maintained GCT exhibits loss fluctuations in the later stages. It indicates that constraints adversely affect model convergence, while a simple decay can effectively resolve it.

The variation in gradient code length shows that decaying GCT maintains a significantly lower and more stable code length than conventional training, especially in the later stages, confirming that the stabilization effect is positively correlated with the Lagrange coefficient's magnitude.

### 5.3 GENERALIZATION

Given that numerous factors like model scale, architecture, and datasets introduce task heterogeneity and directly impact training outcomes, we will validate the generalization of GCT across various models and tasks. Specifically, we will conduct classification tasks on BERT and GPT models using classification accuracy as the metric, and perform generation tasks on LLaMA and DeepSeek models using classification accuracy as the metric.

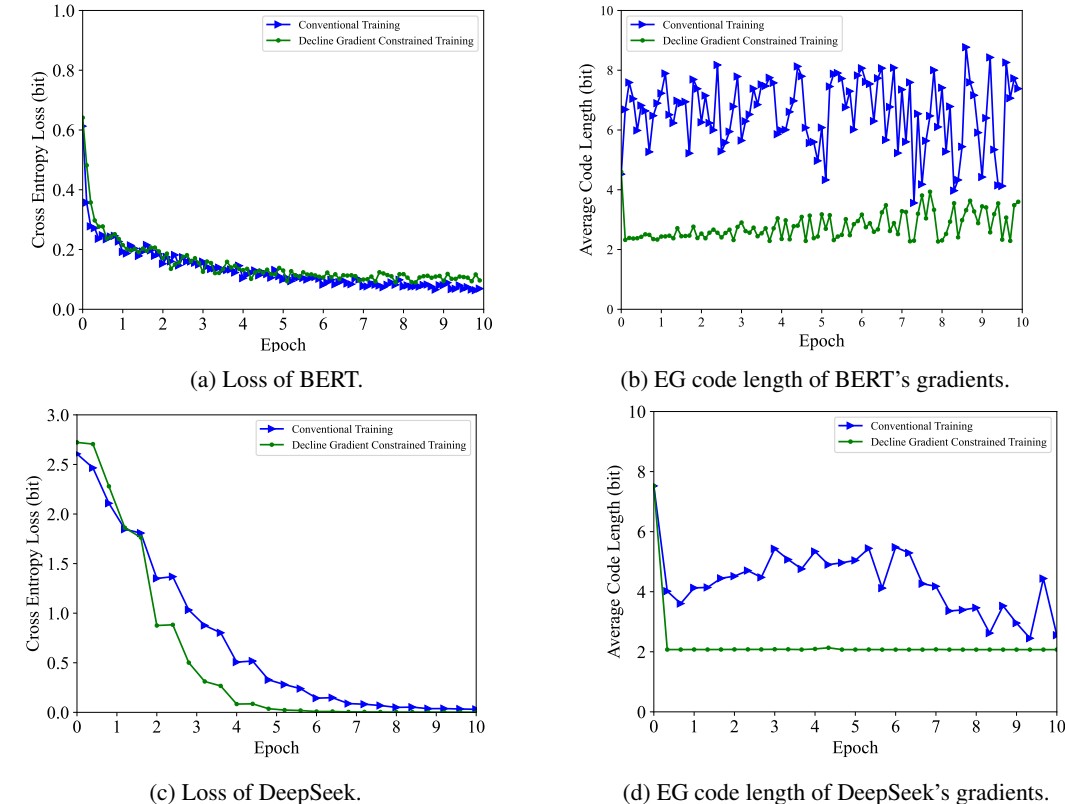

Figure 2: Changes in loss values and EG average code length when BERT and DeepSeek models are trained with conventional training and GCT respectively.

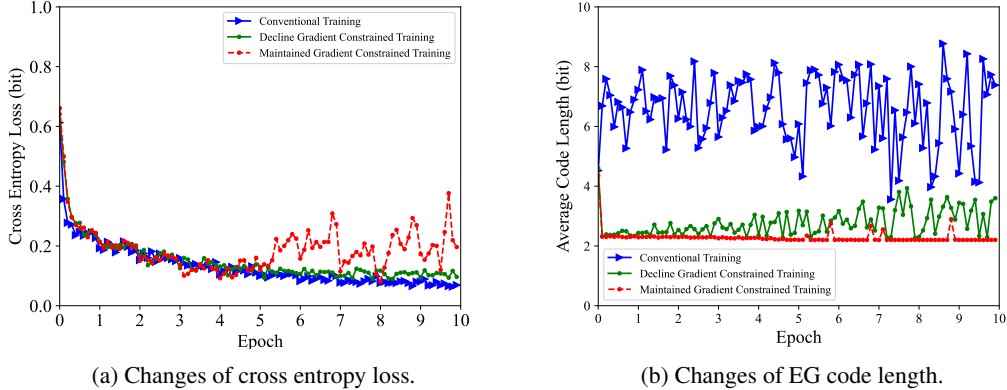

Figure 3: The variation of loss and average code length for GCT without decay, with decayed GCT and conventional training as control groups.

In terms of compression effectiveness, for each model trained with gradient constraints, the average code length of gradients after EG encoding achieved a compression rate of no less than 70%. In terms of model performance, GCT models under most configurations achieved higher accuracy on the test set than conventionally trained models. This phenomenon suggests that, beyond effectively compressing training gradients, the gradient constraint training method may have a potential, positive effect on improving the model's generalization capability.

Table 1: Compression performance of GCT with different model architectures and datasets.

| Model | Param Size | Dataset | Method | EG (bits) | FL (bits) | EG Compress | Accuracy |
|---|---|---|---|---|---|---|---|
| BERT | 110M | IMDB | - | 6.58 | 11.16 | 41% | 92.39% |
|  |  |  | GCT | 2.28 | 10.81 | **79%** | **92.46%** |
| GPT | 774M | IMDB | - | 5.01 | 11.12 | 55% | 95.38% |
|  |  |  | GCT | 2.76 | 11.04 | **75%** | **95.53%** |
| LLaMA | 1B | SQuAD | - | 7.83 | 11.96 | 35% | **99.95%** |
|  |  |  | GCT | 3.02 | 10.36 | **71%** | 98.80% |
| DeepSeek | 7B | SQuAD | - | 4.64 | 11.40 | 59% | 99.97% |
|  |  |  | GCT | 2.30 | 10.80 | **79%** | **99.98%** |

## 5.4 DISTRIBUTED TRAINING INTEGRATION

The demand for gradient compression arises from the broad application of distributed training architectures, making its effectiveness in this setting critical. For validation, we leveraged our custombuilt, EG-coding-based distributed training framework to parallelly train a BERT model on two A100 GPUs. The training results are presented in Fig 4.

The observed trends align with the single-machine results presented in Section 5.1. GCT and conventional training exhibited similar convergence rates, though GCT appeared to settle at a slightly higher loss. Tests confirmed it does not harm the model's generalization capability and accuracy on the test set showed comparable results: 92.51% for GCT and 92.61% for conventional training.

Meanwhile, the gradient code length of GCT remained significantly lower. This result corroborates that the effectiveness of GCT is consistent across different computational architectures (singlemachine and distributed), thereby validating the reliability of this method in distributed environ-

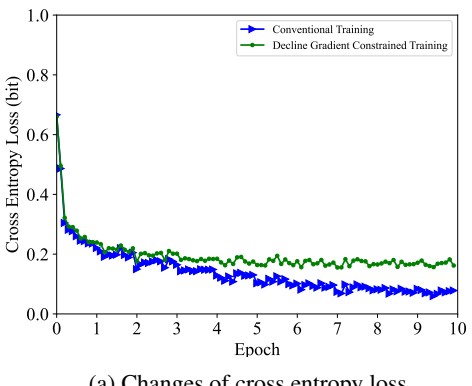

(a) Changes of cross entropy loss.

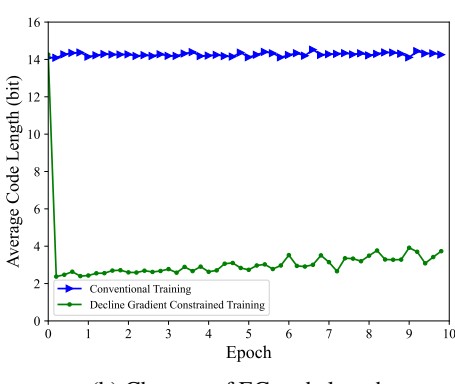

(b) Changes of EG code length.

Figure 4: The variation of loss and average code length for GCT in distributed training. Each step in the graphs represent the average of the loss values and the gradient code lengths from the models on both GPUs.

## 6 CONCLUSION

This paper proposes a novel gradient constrained training method based on rate-distortion optimization. For the first time, this method introduces a gradient constraint term into the loss function to generate highly sparse gradients, thereby reducing the compression difficulty for subsequent algorithms such as quantization, sparsification, and entropy coding. Experimental results demonstrate that, without compromising model training accuracy, the method achieves a gradient compression rate of no less than 70% using EG encoding. The GCT algorithm also exhibits strong generalization and robustness across various tasks and in distributed training. This novel method is likely to have a profound impact on future distributed training and its hardware design.

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

# A  APPENDIX

## A.1  USE OF LLMS

In this research, large language models (LLMs) were used as auxiliary tools for text polishing and selective sentence translation. The AI system assisted in improving the clarity and flow of academic writing, and provided translation support for certain complex sentences from Chinese to English.

All AI-generated content was carefully reviewed and modified by the authors to ensure accuracy and maintain the original research intent. The core ideas, methodology, and conclusions remain the original contributions of the authors.

