# OpenReview forum: "Gradient-Constrained Training for Distributed Large Language Models"
_ICLR.cc/2026/Conference — Submitted to ICLR 2026_

### Official Review · Reviewer_UuRS · 2025-10-27

**Soundness:** 1
**Presentation:** 2
**Contribution:** 1
**Rating:** 2
**Confidence:** 4

**Summary:**

This paper proposes a gradient-constrained training framework that reduces gradient communication. By integrating gradient distribution constraints into the loss function, the method produces sparser, more stable gradients which benefits downstream compression using Exponential-Golomb (EG) coding. The authors also introduce a low-overhead approximation for second-order derivatives to further enhance its performance.

**Strengths:**

The paper tries to provide a practical approach to improving gradient compressibility in distributed training and is easy to follow. While building on established ideas (e.g., rate-distortion theory and gradient sparsity), it combines gradient-constrained optimization with EG coding and a lightweight second-order approximation.

**Weaknesses:**

1. Novelty concerns.
    a. The ideas are not quite new. For example, [1] proposed a more well-sounded pipeline for gradient compression in distributed training, which includes a lossless encoding stage. And the GCT target emerges naturally from RDO theory.
    b. Using gradient difference for second-order derivatives is already presented in previous work, as in [2].

2. Insufficient experiments.
    a. The author selects EG encoding over traditional encoding methods, but there's no comparison in experiments.
    b. Training speed improvement under low-bandwidth environment is not provided, which is important to understand how much gradient compression contributes to the training.
    c. The evaluation task selected are saturated (all 90%+, even at 99%). For example, GCT gets a higher loss in Figure 4, but comparable accuracy to the baseline. Across the four experiments in Table 1, the improvement lacks confidence and the claim that GCT improves the model’s generalization capability is counterintuitive, because of its modified learning target.

Reference:
[1] COMPSO: Optimizing Gradient Compression for Distributed Training with Second-Order Optimizers
[2] AGD: an Auto-switchable Optimizer using Stepwise Gradient Difference for Preconditioning Matrix

**Questions:**

1. In Section 3.1, the paper says, *Our systematic analysis reveals that gradients across various models and training stages consistently exhibit peaky, heavy-tailed distributions, well-characterized by generalized Gaussian distributions with low shape parameters*. However, the paper only briefly describes the gradient distribution during BERT training. Does this phenomenon extend to the training of other models? For reference, I find a post about the training dynamics of Pythia 125m model [1], the gradients *begin as Gaussian, slowly shrink in variance with expanding tails until, at about 20,000 steps, they undergo a sudden shift to a logistic distribution*.
2. In Table 1, does the DeepSeek model refers to the `deepseek-ai/DeepSeek-R1-Distill-Qwen-7B`? It would be better to refer the model more accurately throughout the paper.
3. Provide more experiments setup (eg. training framework, implementation details, hyper-parameter settings) of the experiments for reproduction.

Reference:
[1] https://www.lesswrong.com/posts/2JJtxitp6nqu6ffak/basic-facts-about-language-models-during-training-1#So_do_residual_stream_and_weight_gradients

---

### Official Review · Reviewer_xDVx · 2025-10-29

**Soundness:** 1
**Presentation:** 2
**Contribution:** 1
**Rating:** 2
**Confidence:** 4

**Summary:**

The paper proposes a new gradient update algorithm called GCT, which, after applying Exponential Golomb (EG) encoding, achieves significantly higher compression efficiency compared to traditional gradient update methods with EG encoding, thereby reducing gradient communication overhead.

**Strengths:**

The authors attempt to propose a new gradient update algorithm that achieves higher compression efficiency under a specific encoding scheme; the idea itself is quite interesting.

**Weaknesses:**

- The authors claim that "our systematic analysis reveals that gradients across various models and training stages consistently exhibit peaky, heavy-tailed distributions, well-characterized by generalized Gaussian distributions with low shape parameters." However, the paper only presents gradient distribution results for BERT on the IMDB dataset, lacking evidence of systematic analysis across diverse models and datasets as implied.
- Could the authors clarify the relationship between $\hat{h}_i^{(t)}$ and $h_i^{(t)}$? The theoretical justification for approximating $\hat{h}_i^{(t)}$ rather than $h_i^{(t)}$ requires further explanation.
- There appears to be a notation issue in Equation 6 - the right-hand side should likely be $-\eta{}\cdot\hat{g}_i^{(t-1)}$ based on the context.
- The convergence analysis for GCT (Gradient Compression Technique) is notably absent from the theoretical framework.
- The experimental section would benefit from more detailed documentation, including: optimizer specifications, hyperparameter configurations, and whether hyperparameter optimization was performed.
- In line 362, should $\lambda$ perhaps be $\alpha$ instead?
- A comparative analysis of actual runtime performance would strengthen the practical evaluation of the proposed method.
- The study lacks experiments involving training-from-scratch scenarios to better isolate the method's effectiveness.
- The distributed training validation seems limited in scale — expansion to multi-node, multi-GPU environments would provide more comprehensive evidence for scalability.

**Questions:**

Please refer to weeknesses.

---

### Official Review · Reviewer_iALt · 2025-10-31

**Soundness:** 3
**Presentation:** 3
**Contribution:** 2
**Rating:** 4
**Confidence:** 3

**Summary:**

This paper addresses communication-efficient training by proposing Gradient Constrained Training (GCT), a method that uses gradient constraints to generate low-rate gradients. The core idea is to apply a regularization term during training, thereby inducing sparsity in the gradients.

**Strengths:**

The paper is well-written, with a clearly articulated motivation and a well-structured presentation of the proposed method.

**Weaknesses:**

I have several major concerns regarding the methodological soundness and specialization of this work:

1. Limited Expressivity: The introduced regularization, while promoting sparsity, may significantly constrain the model's expressivity. This could lead to suboptimal performance on complex tasks, as the network's capacity is artificially limited. For instance, what is the final training loss achieved for the GPT-small model, and how does it compare to the baseline?

2. Optimization Difficulty: The proposed regularizer is non-smooth, which inherently complicates the optimization process. The paper does not adequately address how this challenge is mitigated, potentially leading to unstable or inefficient training.

3. Lack of Architectural Specialization: The method appears to be a generic application of regularization and does not incorporate any design elements specifically tailored for Transformer-based LLM architectures, which limits its novelty and potential effectiveness in this domain.

**Questions:**

1. What is the final training loss achieved by the proposed method on the GPT-small model? This is crucial information for assessing its convergence performance.

2. To properly evaluate training efficiency, could you provide a plot of the training loss against wall-clock time, comparing your method with the baseline approaches?

3. Were any additional techniques, such as gradient clipping or a reduced step size, required to ensure stable optimization with the proposed regularization?

4. Does the implementation include any specific modifications or treatments that are tailored to the architecture of Large Language Models (LLMs), such as the Transformer?

---

### Official Review · Reviewer_joKB · 2025-10-31

**Soundness:** 2
**Presentation:** 1
**Contribution:** 1
**Rating:** 2
**Confidence:** 5

**Summary:**

This paper introduces Gradient-Constrained Training (GCT), a novel method for communication-efficient distributed LLM training that aims to generate inherently sparse gradients directly, rather than compressing them post-computation. The core idea is to reframe the optimization as a rate-distortion problem by adding a gradient constraint term to the main task loss. This constraint encourages the training process to produce gradients that are "compression-friendly" and exhibit a sparser, generalized Gaussian distribution, making them highly compressible with entropy coding like Exponential Golomb (EG) coding. To make this tractable, the authors propose an efficient quasi-Newton approximation to compute the required second-order derivatives with minimal overhead. Experiments show that GCT can achieve average bitrate savings during training without degrading, and in some cases even improving, model convergence and final accuracy.

**Strengths:**

The paper's exploration of gradient distributions (Section 3.1) and its proposal to use Exponential Golomb (EG) coding is a valuable contribution. Identifying that gradients follow a generalized Gaussian distribution and applying a suitable entropy coding scheme is a novel approach to gradient compression that moves beyond standard sparsification or quantization.

**Weaknesses:**

1. Many of the paper's claims lack sufficient evidence, leading to a weak motivation for the core idea. The proposed method, GCT, appears less like a formal constrained optimization problem and more like a heuristic regularization term designed to achieve stability when combined with the proposed Exponential Golomb (EG) coding. The authors do not provide references or a theoretical derivation in Section 4.1 to justify the choice of the gradient's $l_1$ norm as the rate-distortion "rate" term. Furthermore, the paper lacks any theoretical analysis of how this $l_1$ regularization term impacts convergence. The authors' intuition remains unclear, as the discussion heavily favors the efficient computation of the second-order approximation (Section 4.3) rather than the fundamental motivation for the regularization term itself.

2. The experimental validation is overly simplistic and unconvincing. The paper is entirely missing pre-training experiments, which are a primary use case for large-scale distributed training and communication optimization. The fine-tuning experiments provided are also superficial, lacking a strong selection of modern baseline models and proper validation across a diverse suite of downstream tasks. Simply fine-tuning BERT on IMDB is insufficient to prove the method's effectiveness in a genuinely challenging optimization scenario. The fine-tuning loss curves are not a strong indicator of optimization capability, as the pre-trained model already starts near a stable local optimum. In Figure 2 (b), the proposed algorithm does not converge.

3. The overall presentation of the paper has significant issues. First, it lacks clear definitions for many of the symbols used, nor does it provide formal assumptions, propositions, or theorems to justify the algorithm's core steps. The parentheses are not in the correct format. The experimental figures also have presentation flaws; for instance, in Figure 2, the fourth plot (d) is not properly aligned with the other three. Furthermore, the axis representations across figures are inconsistent in font size, decimal precision, and formatting. The related work section is disproportionately large yet fails to clearly connect the cited works to the authors' own contributions, resulting in an imbalanced paper structure.

**Questions:**

I think this paper needs significant revisions to meet the requirements of ICLR.

---

### Meta-Review · Area_Chair_6sou · 2026-01-01

**Summary:**

Clear consensus of rejection and no rebuttal

**Reviewer Concerns:**

N/A - no rebuttal

**Reviewer Scores:**

N/A - no rebuttal

---

### Decision · Program_Chairs · 2026-01-26

Reject